# Digital twin simulation modelling shows that mass testing and local lockdowns effectively controlled COVID-19 in Denmark

Kaare Græsbøll [1,2] ✉, Rasmus Skytte Eriksen [1], Carsten Kirkeby [3] & Lasse Engbo Christiansen [1,2]

## Abstract

**Background** Following the COVID-19 pandemic, it is important to evaluate different mitigation strategies for future preparedness. Mass testing and local lockdowns were employed during the Alpha wave in Denmark, which led to ten times more tests than the typical European member state and incidence-based restrictions at the parish level. This study aims to quantify the effects of these interventions in terms of hospital admissions and societal freedom.

**Methods** This study assesses the effectiveness of these strategies via counterfactual scenarios using a detailed, individual-based simulation model that replicates the entire Danish population. The model considers multiple factors, including evolving societal restrictions, vaccination roll-out, seasonal influences, and varying intensities of PCR and antigen testing across different age groups and degree of completed vaccination. It also integrates adaptive human behavior in response to changes in incidences at the municipality and parish levels.

**Results** The simulations show, that without mass testing in Denmark, there would have been a 150% increase in hospital admissions, and additional local lockdowns equivalent to 21 days of strict national lockdown. Without the policy of local lockdowns, hospitalizations would have increased by 50%.

**Conclusions** In conclusion, the combination of mass testing and local lockdowns likely prevented a large increase in hospitalizations while increasing overall societal freedom during the Alpha wave in Denmark. In future epidemics, mass testing and local lockdowns can likely prevent overwhelming healthcare systems in phases of high transmission and hospitalization risks.

## Plain Language Summary

This study looked at how Denmark handled the COVID-19 pandemic, specifically focusing on mass testing and local lockdowns during the Alpha-wave. Compared to other European countries, Denmark conducted ten times more tests and implemented restrictions at the parish level based on local incidence. Using a detailed simulation model, the researchers explored what would have happened without these measures. Without mass testing, hospital admissions would have increased by 150%, and without local lockdowns, they would have gone up by 50%. Furthermore, mass testing prevented 21 days of strict national lockdown. In essence, mass testing and local lockdowns in Denmark prevented a substantial rise in hospitalizations while allowing more overall societal freedom. This highlights the importance of these strategies for future pandemic preparedness.

Mass testing, defined as "regular asymptomatic screening of the general population", has been proposed as an effective means of controlling disease spread[1]. However, comparing test incidences with case incidences across countries may be uninformative due to inherent differences between countries. As an example, in the first half year of 2021, during the SARS-CoV-2 Alpha wave, Czechia and Cyprus were among the European Union (EU) member states which tested the most per inhabitant but still had some

of the highest incidence rates, while Finland and Iceland tested very little and still had a low incidence rate[2]. To test the actual impact of the mass testing strategy, all mitigating factors implemented within the country must be taken into account. Therefore, we assess the EU member with the highest test incidence during the period: Denmark.

On December 16, 2020, in response to an increasing number of hospital admissions attributable to SARS-CoV-2, the Danish government

[1]Department of Epidemiology Research, Statens Serum Institut, Artillerivej 5, Copenhagen S, 2300, Denmark. [2]Dynamical Systems, Compute, Technical University of Denmark, Anker Engelunds Vej 101, Kongens Lyngby, 2800, Denmark. [3]Department of Veterinary and Animal Sciences, Faculty of Health and Medical Sciences, University of Copenhagen, Grønnegårdsvej 8, Frederiksberg C, Copenhagen, DK-1870, Denmark. ✉e-mail: kgrb@ssi.dk

instituted a strict national lockdown[3]. This lockdown was incrementally eased during the first half of 2021, coinciding with a substantial increase in testing. Part of this increase was due to the introduction of the "Corona passport", which required a negative SARS-CoV-2 test for individuals to participate in certain societal activities[4], and screening tests performed in teaching institutions. In the months of January to June 2021, the volume of individuals tested equated to approximately 9.5 times the total population of Denmark, which is more than ten times the amount in the typical EU member state at the time[2,5]. This extensive testing initiative was supplemented by automated local lockdowns – mechanisms that activated lockdowns in municipalities and parishes that exceeded specific incidence thresholds.

To assess the impact of the chosen strategy, a digital twin of the entire Danish population was created based on register data. The population was stratified by age, vaccination target group, vaccination status, and geography down to the parish level. Progression of time was modeled using an individual-based model where disease transmission occurs via homogeneous mixing on two levels: within municipalities and nationally. Due to the model using methods from both agent-based and compartment models, it can be considered a hybrid model, and it is referred to as the "popIBM" (population Individual-Based Model).

Mass testing is a strategy that aims to catch as many infected individuals as early as possible after infection, to the effect that they can isolate themselves and thus limit the period in which they may infect others. Local lockdowns is a strategy for periods where a disease spread very heterogeneous across geographic entities. This strategy aims to reduce contacts and thus prevent transmission in areas with high transmission, while keeping open society in areas with low transmission. Denmark used 'automatic' local lockdowns, meaning that the criteria for lockdown was publicly know, and thus local authorities and residents could make efforts to reduce transmission when approaching those criteria, in order to try and avoid a lockdown.

Aside from the strategies of mass testing and local lockdowns, at least six other factors substantially impacted the spread of SARS-CoV-2 during the study period (January 11 to June 30 of 2021): 1) The rise of the SARS-CoV-2 variant B.1.1.7 (WHO designated name: Alpha), known for increased transmission risk and a higher probability of hospitalization compared to earlier variants in Denmark[6–8]. 2) The initiation of a vaccination campaign, commencing on December 28, 2020. The Danish vaccination strategy initially targeted the elderly and healthcare workers before expanding to the general population[9,10]. By the end of the study period in July 2021, 26% of the Danish population was fully vaccinated. 3) A stepwise reopening of Danish society in the first half of 2021, in response to an improving epidemic situation. By June 2021, Danish society was nearly free of restrictions. 4) Seasonal variations in transmission. As with other respiratory infectious diseases, the transmission of SARS-CoV-2 has been observed to vary with the seasons[11]. 5) Observable behavioral changes among local authorities and residents when municipalities and parishes approached the automatic lockdown threshold. Local authorities generally introduced soft measures to curb transmission and encouraged residents to undergo testing. 6) Parish-level variation in infection risk. Various factors influenced the transmission risk, leading to heterogeneity in the spread across geographical entities. All of the aforementioned factors were integrated into the model. Notably, the model parameters were not tailored for this specific study; in fact, all but one parameter were estimated by April 2021, and there has been no retrospective optimization of parameters undertaken for this study.

We introduce the concept of an "openness index", which describe the relative openness of society in terms of contacts between individuals. In which the maximal open society has the openness index value 1, and restrictions on contact will reduce this value. The openness index changes step-wise during the study period from 0.47 to 1, as Denmark start in strict nationwide lockdown on January 2021 and to an almost restriction free society in June 2021.

The daily cost of mass testing per inhabitant of Denmark was calculated to be 0.90€ for the first half of 2021, and here we try to quantify what was gained spending this amount.

In this study, we use this model to compare the Danish mitigation strategies with scenarios comparable to those of other European member states. By estimating the relative effect of the Danish strategy during this period, recommendations for future epidemics can be formed.

Simulations show that especially mass testing prevents hospitalisations while increasing societal freedom. We recommend that mass testing and local lockdowns be considered as effective tools to control future epidemics.

## Methods
The popIBM utilized here is an individual-based stochastic model designed to align with the demographic distributions of Denmark. This means the individuals in the model are not matched to specific real-world individuals, but rather the model generates a synthetic population with comparable age, vaccination target group, and place of residence to the Danish population.

### Population
The synthetic population was initialized by assigning each individual to an age group and a vaccination target group, followed by distributing them across municipalities and parishes. This assignment was based on data from the Danish central person registry (CPR), which contains information about the birth year and a unique personal identification number that can be matched with other registries. The population was stratified into nine age groups (0-9, 10-19, …, 80+ years). Each individual belongs to one of the 2100+ parishes, which are nested in the 98 municipalities of Denmark.

By July 2021, the Danish health authorities had prioritized vaccination for 17 distinct target groups. These vaccination target groups represent the prioritized rollout of vaccines in Denmark. This rollout initially targeted the elderly and healthcare personnel, gradually extending by age to 50+ year-olds, after which a pronged approach was used (see the Supplementary Methods A.1 for more details). By the end of the study period, 26% of the Danish population was fully vaccinated. The model effectively has 18 vaccination target groups, as those aged 0-11 years were not eligible for vaccination during the study period, and therefore not included in the official target groups.

Individuals in the simulation belongs to both an age group and a vaccination target group. The age group defines the number of contacts through the contact matrices (see Supplementary Methods A.2 for details), the vaccination target group determines the timing of receiving vaccines (see Supplementary Methods A.1 for details), and the combination of age group and vaccination target group determines the risk of going to hospital (see Supplementary Fig. A2 for details).

For each vaccination target group, vaccines were distributed weekly, as reported in[10]. This was replicated in the model by pre-assigning vaccination dates to all relevant individuals. We assumed that vaccine doses were evenly distributed across the seven days of the week during which they were supplied.

In the model each vaccinated individual had a countdown timer set to the date when the vaccination would take effect; this was 21 days after the first dose given according to the vaccination roll out for the Comirnaty and Spikevax, and 28 days for the Vaxzevria. This period was based on the expectation that individuals would receive their second dose in line with the prevailing guidelines. From our data, we found that 98.3% of people who enrolled in the vaccination program during the study period eventually attained "fully vaccinated" status. In effect, we assume that all vaccinated persons in the model follow the vaccination guideline and obtain the effect of the vaccine immediately after the second dose.

### Initialization
Simulation runs began on Monday, January 4, 2021, and were initialized with 4% of the infected individuals harboring the SARS-CoV-2 Alpha strain, reflecting the situation observed in Denmark at that time[12,13]. We did not initialize the simulation before this date due to the difficulty of accurately

determining activity levels over the holiday period. This was followed by a one-week 'burn-in' period. The study period under consideration was January 11 to June 30, 2021.

## Disease spread

In the model, individuals could occupy one of four disease states, corresponding to the classic SEIR model: susceptible, exposed, infectious, or recovered. Disregarding the effects of local lockdowns, the probability of an individual becoming exposed during a time step was calculated in two phases, in which in- and outgoing transmissions from individuals were separately evaluated:

The first step was to compute outgoing transmission, equivalent to the force of infection $\mathbf{F}$, per age group at both the national and municipal levels. Since some infected individuals were in (self-)isolation due to a positive test or local lockdowns, the effective number of infectious individuals was reduced. The number of infectious people capable of transmitting the disease, $I_{am}^{\text{eff}}$, varied across age groups, $a$, and municipalities, $m$. The force of infection was determined by multiplying the number of effectively infectious individuals with the time-varying activity matrix, $\mathbf{A}(t)$, which outlined activity in Danish society on a national scale (see Supplementary Methods A.2 for details):

$$I_{am}^{\text{eff}} = \sum_i \text{I}(\text{disease}_{ampi} == \text{infectious}) \min(\Delta^m(\text{inc}_m), \Delta^p(\text{inc}_p)) \xi_i \quad (1)$$

$$\mathbf{F}_{\cdot m} = \beta_0 \beta(T) \mathbf{A}(t) \mathbf{I}_{\cdot m}^{\text{eff}} \quad (2)$$

$$\mathbf{F}_{\cdot}^{\text{DK}} = \sum_m \mathbf{F}_{\cdot m} \quad (3)$$

Here $\mathbf{F}_{\cdot m}$ is the per-municipality force of infection vector that includes the nine age groups (implicit $a$) for the $m$'th municipality and $\mathbf{F}_{\cdot}^{\text{DK}}$ is the nation-wide force of infection from each age group. For all parameters, the subscript $i$ refers to the individual. $\beta_0$ is the overall transmission rate, $\beta(T)$ is a temperature-dependent scaling of the transmission rate per contact (see Supplementary Methods A.6.2), and $\mathbf{I}_{\cdot m}^{\text{eff}}$ is the vector of number of effective infectious individuals that includes the nine age groups (implicit $a$ represented by $\cdot$) for the $m$'th municipality. The functions $\Delta^m(\text{inc}_m)$ and $\Delta^p(\text{inc}_p)$ describe the effects of local lockdown in municipality $m$ and parish $p$ respectively as a function of incidence in the entity (see Supplementary Methods A.6.1 for details), and $\xi_i$ represents individuals self-isolating at home due to a positive test. When an individual is not in local lockdown or self-isolated due to a positive test the $\Delta^x(\text{inc}_x)$ and $\xi_i$ parameters take a value of 1. They take a value in the range $[0 : 1[$ if an individual is in an area under local lockdown or if they are in self-isolation due to a positive test. Please note that national lockdown measures are included in the $\mathbf{A}(t)$ matrix.

When the incidence of the previous week in a given period exceeds a specific threshold the area will enter into a lockdown similar to the strict lockdown of January 2021 (one threshold for municipalities and another, higher threshold for parishes. Lockdowns would be lifted if the area had been consistently under the limit for seven days[14]. Thresholds were incrementally increased throughout the study period. See Supplementary Methods A.6.1 for details). This results in both in-going and out-going contacts being reduced. These reductions are approximated by modifying the probability of becoming exposed or transmitting the disease by a factor of $\delta(t) = \sqrt{\lambda(0)/\lambda(t)}$, where $\lambda$ represent the dominant eigenvalue of the activity matrix, $\mathbf{A}(t)$, at times $t$ and $t = 0$. The latter corresponds to the beginning of the study period under strict lockdown. Note that instead of counting every infectious individual with weight 1, individuals under lockdown will be weighted by $\Delta^x(\text{inc}_x)$. This modification of in-going and out-going contacts has the asymptotic behavior, that if everybody is under local lockdown at any given time, $t$, the force of infection is approximately the same as at $t = 0$. It does not match exactly, due to the lockdown at $t = 0$ having a different impact across age groups than the re-scaled lockdowns at times $t$. Due to the introduction of soft restrictions when municipalities and

parishes were nearing the incidence limits, the $\Delta^x(\text{inc}_x)$ functions would change in a piecewise linear way towards $\delta(t)$ (see Supplementary Methods A.6.1 for details).

Test-positive individuals isolate themselves which set their outgoing contacts to $\xi_i = 0$. In reality, this number is likely higher than zero, as not all individuals were expected to fully comply with guidelines. However, there was generally high compliance in Danish society, and free accommodation was offered by the Danish government for any person not able to isolate at home.

Combined, the probability for an individual $i$ of moving from a susceptible to an exposed state at each time step, $P_i(S \to E)$, depends on the local, age-group dependent probability $P_{am}(S \to E)$ modified by the vaccination and lockdown status of the individual. Mathematically, this probability becomes:

$$P_{am}(S \to E) = 1 - \exp\left(-(1-\alpha)\frac{F_a^{\text{DK}}}{N_a^{\text{DK}}} - \alpha\frac{F_{am}}{N_{am}}\right) \quad (4)$$

$$P_i(S \to E) = P_{am}(S \to E) \min(\Delta^m(\text{inc}_m), \Delta^p(\text{inc}_p)) V_{ki} \rho_p \quad (5)$$

where $\alpha$ is the fraction of transmission going on at the municipality level (throughout this study set to 90%), and $N_{am}$ is the population of each age group $a$ in each municipality $m$, with $N_a^{\text{DK}} = \sum_m N_{am}$. The parameter $V_{ki}$ represents an individual factor describing whether an individual has been fully vaccinated with vaccine type $k$.

Furthermore, each parish, $p$, has a risk factor, $\rho_p$, in the model. This factor is based on the observed cumulative incidence in parishes in Denmark up til the beginning of the study period (see Supplementary Methods A.3.1). This parish-level risk factor has the range of $[0.49; 1.57]$ and is a proxy for many unobserved risk factors within the parish (i.e. population density, socioeconomic status, etc). The parish risk factor was introduced in the model to match the observed spatial heterogeneity in the epidemic. If parish risk factors are not introduced, incidences will be very homogeneous across geographic entities.

For the SARS-CoV-2 Alpha variant, the vaccine effectiveness were high[15], and thus, in the model, the probability of being infected is reduced by a vaccine-dependent factor $V_k$. This factor is assumed to be 90% 21 days following the first dose of the Comirnaty and Spikevax vaccines and 60% 28 days following the first dose of Vaxzevria vaccination (see Supplementary Methods A.5 for details). The delay until effect is based on the assumption that individuals receive the second dose according to the vaccination guidelines available at the time.

Note that the above equations assume that no one self-isolates due to a false-positive SARS-CoV-2 test (due to high test specificity[16]).

When individuals enter the exposed and infectious stages, the time to move to the next stage (respectively infectious and recovered) is drawn from a gamma distribution with shape parameter $k = 2$ and a mean time $1/\gamma_E = 2.5$ days and $1/\gamma_I = 5.3$ days respectively. Furthermore, 50% of exposed individuals (based on estimates by Johansson et al.[17]) will draw a time until onset of symptoms also from a $k = 2$ gamma distribution $1/\gamma_{\text{symp}} = 2$ days. Upon symptom onset, individuals will be tested and isolate themselves.

In addition, the general population and infected without symptoms are tested randomly following the observed time-varying test incidence stratified by age and vaccination status. The tests were furthermore spatially distributed according to the time-varying incidence in the municipality so that areas with high incidence would be tested more – in accordance with observed behavior (see Supplementary Methods A.3.2 for details).

## Hospitalization

The risk of being admitted to a hospital follows the observed risk in the period up to the study period stratified by age and vaccination target group and then multiplied with the observed risk factor of 1.42 for the Alpha variant[7]. This way, the risk is representative of Danish practices and likely also how these groups have contact with the health system and the general health status of these groups. There is not taken into consideration further

factors such as specific comorbidities for individuals in these target groups. Vaccination also reduces the risk of going to hospital independently of reducing the risk of getting infected. See Supplementary Table A2 and Supplementary Fig. A2 for details.

## Societal openness

During the study period, Denmark went from a strict national lockdown in January 2021 to a very open society by July 2021. To describe this in simple terms, it was chosen to use the dominant eigenvector, $\lambda(t)$, of the activity matrix, $\mathbf{A}(t)$, as a proxy for openness in the Danish society, as all national level restrictions are incorporated into the activity matrix. Openness is then defined relative to summer 2021, and the lockdown days reported are relative to the openness of society in summer 2021. Lockdown days reported in the results are relative to the societal activity as it was in the summer of 2021. Using this definition, one lockdown day in summer 2021 where no restrictions are in place would reduce the openness of society as much as roughly two lockdown days in the middle of April were some restrictions are still in place.

## Seasonal effect

To investigate the seasonal effect $\beta(T)$, a SEIR model was fitted on hospital admission in 18 regions of Sweden in the period of April 2020 to October 2020. Sweden was chosen as case study, since in this period it had minimal changes in mitigation strategy, and because Sweden is assumed to have societal behavior similar to Denmark (see Supplementary Methods A.6.2 for details). Later, a similar model fitted to Denmark directly showed a similar shape and size of effect[18] as the model fitted to Sweden.

## Tests for COVID-19

The weekly number of tests in EU member states was downloaded from European Centre for Disease Prevention and Control's webpage[2]. The number of tests was aggregated for weeks 1 through 26 of 2021 (until July 4, 2021), at the country level and divided with population size. Austria, Belgium, Hungary, Latvia, and Portugal had incomplete observations and were excluded from the study. The mean and median of the total number of tests divided by population were calculated excluding Denmark. The ratio of the number of tests per inhabitant performed in Denmark compared to the other EU member states was 13 to the median and 8 to the mean. It was therefore decided that the limited test scenarios should reduce the number of tests by a factor of 10 compared to the actual performed number of tests.

The cost of testing is estimated by using a price of 16.1 EUR (120 DKK) per PCR test and 20.2 EUR (150 DKK) per antigen test, which is not an official price but estimated by the Danish national broadcaster (DR) on basis of information from various Danish authorities[19]. Both figures are rounded to the nearest 10 DKK, the DKK/EUR conversion rate used throughout this paper is 7.44, which is the euro-cent rounded average of the rate during 2021. The price is the total price including salaries for public employed workers, and overhead for private companies. In Denmark, almost all PCR tests conducted were done by the public sector, while most antigen tests were performed by the private sector. Prices per inhabitant were then found by multiplying by the number of tests in the scenarios and dividing by the number of inhabitants and days in the study period.

Mass testing has the effect in the model that infected individuals may be found prior to being symptomatic with the disease, with a probability related to the test frequency in the group which the individual belong, hereafter they are isolated and cannot transmit the disease. In the 'Limited test' scenarios at least 50% of cases are still found due to being symptomatic, but they will typically be found after having transmitted for longer periods. Local lockdowns reduce to varying degree in- and out-going contacts of all individuals belonging to a geographic entity as a function of the proximity of the observed case incidence to the limits. Local lockdowns last at least seven days, but even when lifted high observed case incidences may still reduce

activity in the entity (See Supplementary Methods A.6.1 for details on incidence limits).

Supplementary Methods contains Supplementary Figs. A1–A10 and Supplementary Tables A1–A4.

## Reporting summary

Further information on research design is available in the Nature Portfolio Reporting Summary linked to this article.

## Results

Four scenarios were modeled using the popIBM. Firstly, a baseline scenario that corresponds to the actual implemented strategy in Denmark, the "Mass test, local lockdowns" scenario, which includes the mass testing and the automated local lockdowns. Furthermore, three counterfactual scenarios in which either the number of tests was limited to 10% of what was actually performed and/or the strategy of automated local lockdowns were not implemented – these scenarios are named with "limited test" and "no local lockdowns" to highlight their deviation from the baseline scenario. Note that the "limited test" scenarios perform a comparable number of tests to the typical EU member state.

## Hospitalizations

In Fig. 1a, the daily hospital admissions from the four scenarios are depicted. The baseline scenario ("Mass test, local lockdowns" - illustrated in green in Fig. 1) is a model of the actual situation during the study period (black line). Of all the scenarios, this one had the lowest number of daily admissions at any point. The "Mass test, no local lockdown" scenario (purple in Fig. 1) mirrored the baseline in terms of daily admissions until April when a rise in admissions was observed due to local lockdowns not being introduced in this scenario. The "Limited tests, local lockdowns" scenario (orange in Fig. 1) resulted in more daily admissions relative to the baseline scenario throughout the study period but flattened out due to local lockdowns activating in many geographic entities. The fourth scenario, the "Limited tests, no local lockdowns" scenario (pink in Fig. 1), saw the highest number of daily admissions, consistently exceeding the other scenarios throughout the study period and peaking with more than 200 admissions per million inhabitants per day during May.

Note that the range reported in parentheses following results is not indicative of a confidence interval but rather of the range of the results across 100 simulations. In other words, these ranges primarily reflect stochastic variation within the model.

Comparing the actual number of daily hospitalizations (black line in Fig. 1a) with the "Mass test, local lockdowns" scenario (green in Fig. 1a), the median cumulative difference over the study period is -65 hospital admissions (range: [-210; 73]). The "Limited test, local lockdowns" scenario (orange in Fig. 1a) resulted in an increase of 1400 per million inhabitants (range: [1200; 1600]), which is an increase of 150% compared to the observed hospitalizations in Denmark (black line in Fig. 1a). The "Mass test, no local lockdowns" scenario (purple in Fig. 1a) resulted in an increase of 470 per million inhabitants (range: [160; 790]), which is an increase of 50% (range: [17; 84]) compared to the observed. The "Limited test, no local lockdowns" scenario (pink in Fig. 1a) resulted in an increase of 10,000 per million inhabitants (range: [9000; 11,000]), which is an increase of 1100% compared to the observed. The results are reported in full in Table 1.

## Societal openness

In Fig. 1b, we display the societal openness during the study period for the four different scenarios. "Openness" is a measure of how unrestricted the contacts between individuals is. The degree of societal openness was calculated by comparing the relative societal activity from the activity matrices in the study period (refer to Supplementary Methods A.2 for details). Formally, the openness index was defined as the dominant eigenvalue relative to the dominant eigenvalue of the contact matrices on June 14 (corresponding

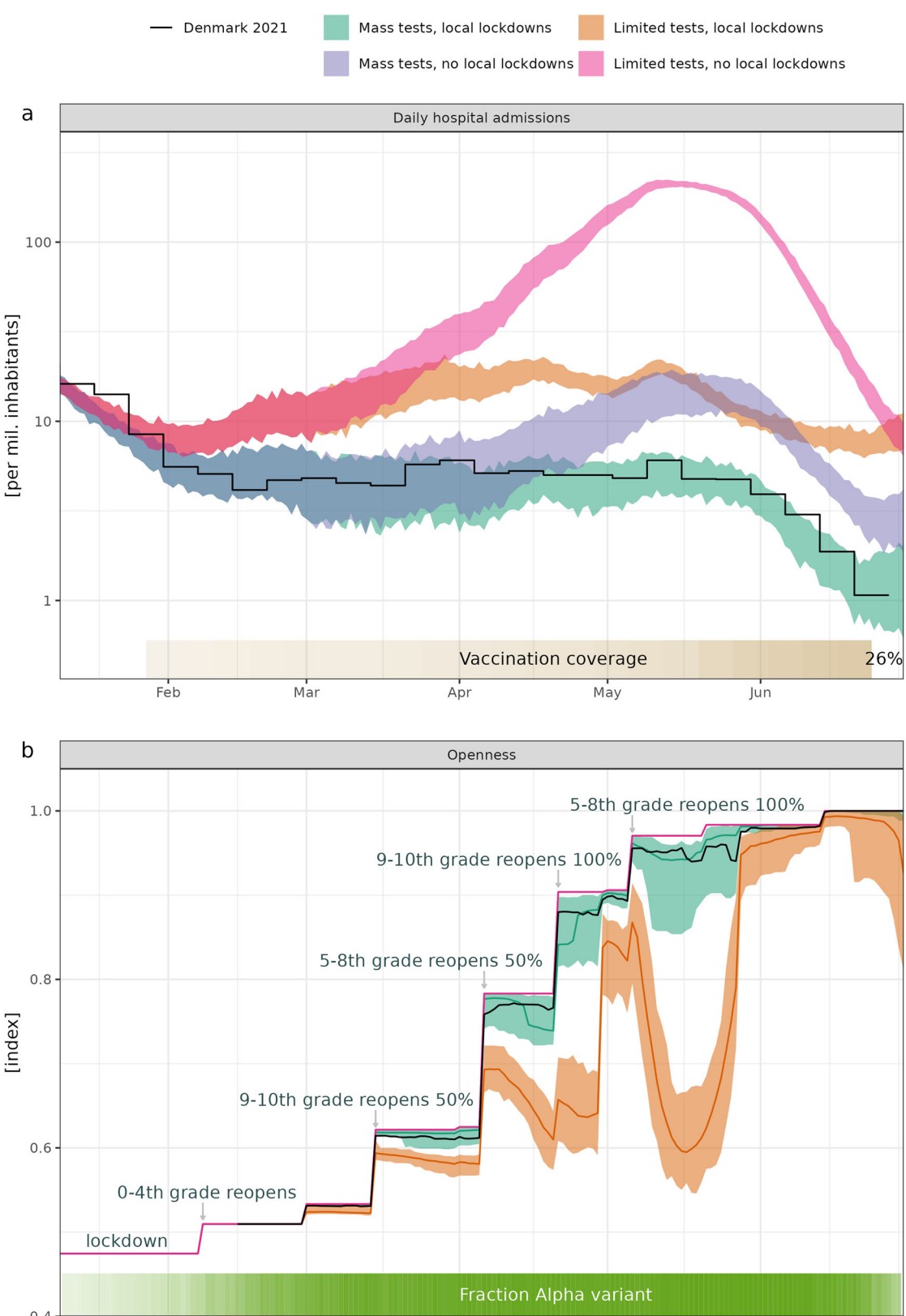

to an almost fully open society). According to this metric, the strict lock-down of Denmark in January 2021 reduced societal activity/openness by 53% (from 1 to 0.47 in Fig. 1b). The impact on the openness of a scenario can be defined as the integral between the black line (observed openness) and the openness of individual runs.

We illustrate the concepts of "openness" and "lockdown days" with some examples: On April 15, the openness index would have been 0.78 if no local lockdowns were present (pink line in Fig. 1b). This means that society would have been 78% open compared to the fully opened society in mid June. However, some local lockdowns where in place which lowered the

**Fig. 1 | Daily hospital admissions and societal openness in four simulated scenarios. a** The number of daily admissions to hospital. Observed development (solid black line) is shown as weekly averages to avoid weekend effects. The four scenarios, shown as colored bands, indicating the minimum and maximum of 100 simulation runs. At the bottom, the percentage of fully-vaccinated individuals is shown in color, ranging from 0% at the start of the study period to 26% at the end of the study period. **b** The degree of societal openness. Actual societal openness is shown as the solid black line. The colored bands indicate simulation results from the four scenarios with the median societal openness of 100 simulations highlighted. At the bottom, the observed fraction of SARS-CoV-2 Alpha cases are shown in green ranging from 0% (transparent) to near 100% (opaque), by the end of the study period it decreases as it is being replaced by Delta. Annotations only denote changes in school openings, but several other changes in restrictions have occurred during this period. For a comprehensive list of changes in restrictions, refer to[33]. Note that openness starts at 0.47 with strict nationwide lockdown in January and goes to 1 in mid June, a reduction of 53%.

**Table 1 | Lockdown days and hospital admissions relative to observed in Denmark January 11 to June 30 2021**

| Scenario | Lockdown [days] | Cumulative hosp. [1 / mil. inh.] | Change hosp. [%] |
|---|---|---|---|
| Observed | Ref. | Ref. | Ref. |
| Mass tests, local lockdowns | 0.00 [−1.3; 1.6] | −65 [−210; 73] | −6.9 [−22; 7.7] |
| Mass tests, no local lockdowns | −2.6 | 470 [160; 790] | 50 [17; 84] |
| Limited tests, local lockdowns | 21 [20; 23] | 1400 [1200; 1600] | 150 [130; 170] |
| Limited tests, no local lockdowns | −2.6 | 10,000 [9900; 11,000] | 1100 [1000; 1200] |

Numbers are medians, square brackets denotes range.
*inh.* inhabitants, *mil.* million, *hosp.* hospitalizations.

openness index to 0.76 (the black line Fig. 1b). The orange line indicates that had there been no "mass testing", openness would only be 0.65 compared to mid June according to our model. All reductions in openness in the black, orange, and green lines compared to the pink are due to local lockdowns. The reduction in openness is proportional to the proportion of affected inhabitants. Had all inhabitants been under local lockdown on April 15, the openness index would have been 0.47, which is identical to the openness during the strict nationwide lockdown in January. If only the greater Copenhagen area (which represent approximately 20% of the inhabitants in Denmark) were under local lockdown, the openness index would be 0.78 - 20% (0.78 - 0.47) = 0.72, which is 20% of the difference between the national restriction level at the time to a strict nationwide lockdown in January. One day of strict national lockdown compared to full openness in mid June corresponds to a change in openness index of 0.53 (distance from 1 to 0.47). The closing of the greater Copenhagen area on April 15 corresponded to a change in openness of 20% (0.78 - 0.47) = 0.06, equivalent to 0.06/0.53 = 0.12 days of national lockdown from fully open to fully closed. The reported results on lockdown are all the integrated difference in openness between the black line and the chosen scenario converted to full lockdown days.

As expected, the baseline "Mass test, local lockdowns" scenario (green in Fig. 1b) aligns with the observed societal openness (the difference of openness is 0.00 (range: [-0.70; 0.88])). The "Limited tests, local lockdowns" scenario in Fig. 1b has a difference in openness of 11 (range: [10; 12]). Given that the strict lockdown reduced openness by 53%, this is equivalent to the "Limited tests, local lockdowns" strategy in Denmark would have resulted in an increase of 21 days (range: [20; 23]) of nationwide strict lockdown (11/53%=21). Finally, the two scenarios without local lockdowns both have slightly higher openness than the observed openness. Over the study period, this sums to 2.6 days of nationwide strict lockdown, meaning that the adopted strategy of local lockdowns in Denmark has resulted in the equivalent of 2.6 days of nationwide strict lockdown.

In Table 1, we display the differences between the scenario results and observed societal openness (in terms of lockdown days) and hospital admissions (in terms of cumulative hospitalizations during the study period). Results comparing observed data with the baseline stratified on age and vaccination status can be seen in Supplementary Fig. A9 and Supplementary Fig. A10.

### Sensitivity analysis
The results of a limited sensitivity analysis showed that it is possible to recreate the epidemic curves of hospitalizations and openness using different parameter combinations, while the results of hospitalizations and lockdown days are either constant or conservative compared to the primary results (See Supplementary Methods A.7 for details).

### Discussion
Estimating the effect of control strategies against COVID-19 across borders is difficult due to many distinct factors (geographic differences in demographic, wealth, vaccination uptake, climate, level of immunity, etc). Therefore, in this study, we estimate the relative effects of two specific strategies within one country, so that they may be compared fairly against each other. The strategies were uniquely implemented in Denmark but could have been implemented in other countries. Our results indicate that mass testing can be used as a relatively inexpensive and effective strategy to reduce hospitalizations without reducing individual freedom in most geographic areas.

Each of the three counterfactual scenarios presented has unique issues as they are, by nature, hypothetical. Thus, it is essential to understand the specific caveats and implications of each.

Firstly, the "Limited test, no local lockdown" scenario forecasts a higher number of hospitalizations than the estimated Danish capacity. Therefore, it's likely that restrictions would not have been lifted as observed in this scenario. From this perspective, it's more useful to conclude that further restrictions were necessary to control the epidemic under this mitigation strategy than to focus on quantitative results.

Secondly, the "Mass test, no local lockdown" scenario shows that omitting local lockdowns increases hospitalizations by 50% during the study period. However, part of the motivation for people getting tested was to avoid local lockdowns. This correlation is included in the model, thereby strengthening local mitigation even without local lockdowns. Investigating this correlation prior to local lockdowns, it can be observed that this drive for tests is smaller – and thus, the scenario might have shown an even higher increase in hospitalizations (see Supplementary Methods A.3.2). It should also be noted that local lockdowns are only in effect for part of the study period (March - June). If looking only at this period, abandoning local lockdowns would have resulted in more than double the number of hospitalizations compared to the observed.

Lastly, the "Limited test, local lockdown" scenario almost entirely controls the epidemic with local lockdowns. The widespread use of local lockdowns during the period is almost equivalent to a national lockdown (the orange line almost reaches the same level of openness as the lockdown in January), and it is reasonable to assume that a national lockdown would

have been implemented instead of local lockdowns. The model implements local lockdowns by effectively re-scaling the contact matrices. This means that the dominant eigenvalue of the contact matrix during nation-wide local lockdowns is identical to that corresponding to the national lockdown in January 2021. However, the modeled local lockdowns' effects are spread evenly across age groups, whereas in reality, local lockdowns disproportionately affected younger people and children. This means that the eigenvectors of the two cases are not aligned. In turn, this discrepancy has dissimilar effects but may have overall similar results, as SARS-CoV-2 transmission is more frequent in younger individuals, while hospitalizations are more frequent in the older. Because this scenario keeps hospitalizations below a level manageable by the Danish health care system at the time, using only methods imposed also in the reference scenario (green), we conclude that the results can be used quantitatively.

It was a primary principle during the model's design to fit the fewest parameters possible within the model, as opposed to using parameters based on external fits (e.g. the seasonal effect, $\beta(T)$). However, parameters such as the balance of spread between the municipality and national level, $\alpha$, and the level of self-isolation following symptoms or a positive test, $\xi$, are fitted within the model. These estimates may impact the model's conclusions. To address this, a sensitivity analysis was conducted to reparameterize the model by adjusting the length of the infectious period by $\pm 2$ days. This analysis found that even if the infectious period was reduced to 3.3 days, the prevented number of strict nationwide lockdown days was comparable to our main findings (see Supplementary Methods A.7 for details).

During the study period, two additional factors affecting the effective infectious period were manual and digital contact tracing. Every individual testing positive for SARS-CoV-2 would be contacted by phone by authorities for contact tracing and testing. Additionally, a contact tracing mobile app was available in Denmark[20]. In this study, we assume the effect of contact tracing is embedded in the parameter for the effective infectious period and is constant over the study period. The Danish national broadcaster (DR) reported that contact tracing success varied between reaching from 2.4 to 2.7 close contacts from high intensity months of January and February to low intensity month of March 2021[21].

The simulation mirrors the actual timing of lifting the national restrictions. If the counterfactual scenarios had played out in real life, the timing of these liftings would likely have been different, as the level of epidemic control was different. Instead of speculating on how the timing would differ, we chose to keep these timings, even though this may have led to worse epidemic control in the counterfactual scenarios. However, it provides a more common basis for comparison between scenarios.

The incidence limits for local lockdowns were raised by the authorities when the epidemic seemed under control (Supplementary Fig. A6). These changes likely would have occurred at different times if hospitalizations had been at different levels, as seen in the counterfactual scenarios. This can be seen, for example, at the end of April when many local lockdowns were lifted (Fig. 1b, orange), almost solely due to changes in thresholds in local lockdown policy (Supplementary Fig. A6). Following this, an increase in hospitalizations is observed, and local lockdowns are again activated across a large part of Denmark.

Adherence by the public to the policies by the authorities is mostly reflected in the data. For mass testing adherence is simply reflected in the number of tests performed, and thus the results should be transferable to other countries. For the effect of local lockdowns it is noted that local lockdowns are assumed to have a similar effect as a strict nationwide lockdown (albeit on a local scale), the effect of the strict nationwide lockdown is estimated based on restrictions (schools, shops, etc.) and an estimate of the reduction in social contacts, which was then fitted by the only free parameter in the simulations, $\beta_0$, which is the overall transmission risk. Other countries may have different effects of lockdowns due to difference adherence. The effectiveness of local lockdowns in other countries than Denmark may thus depend on the overall transmission risk, the heterogeneity of transmission risk, and the overall strength of lockdowns, as well as

the specific incidence limits employed. The incidence limits in Denmark was partly designed in a way so that the resulting hospitalizations would be manageable by the health system, which is also what is observed in the "Limited tests, local lockdowns" scenario. Therefore, employing effective local lockdowns in a future scenario requires a consideration of how observed case incidence relates to hospitalizations.

The model was developed from March to November 2021 by an expert group mandated by the Danish authorities. Consequently, many of the parameters were estimated at different times during this period. The seasonal function, $\beta(T)$, was established in March 2021 and used retrospectively in popIBM and other models by the expert group[22]. All other parameters, except for the $\Delta^x$ function describing the effect of local lockdowns and the adaptive behavior model, were established during March and April 2021[23,24]. The $\Delta^x$ function and the adaptive behavior model were finalized in November 2021[25]. The scenarios tested in this work were devised in November 2022. We believe that not changing parameters after formulating the work's working hypothesis strengthens the conclusions.

According to the simulations, the strategy of mass testing, implemented as a standalone measure, saved Denmark from at least 21 days of strict national lockdown and prevented hospitalization of 1,400 individuals per million inhabitants from January 11 to June 30, 2021, a prevented increase of 150% compared to the observed. Additionally, local lockdowns averted further hospitalization of 470 individuals per million inhabitants, a prevented increase of 50%. When combined, mass testing and local lockdowns prevented an even lengthier strict national lockdown and substantially reduced the number of hospitalizations. The additional cost of mass testing, compared to limited testing scenarios, was approximately 0.81€ per inhabitant per day during the study period. While the absolute numbers are specific to Denmark, the relative effect of these control measures provides an indication that these measures could be effectively used in most countries should a new pandemic break out.

## Data availability
Data on the number of test positive and hospitalizations is publicly available without restrictions from Statens Serum Institut's webpage[26,27]. Source data underlying Fig. 1 is supplied as Supplementary Data 1. All other data are available from the corresponding author on reasonable request, and following legal review whether data can be shared while conforming to the EU's General Data Protection Regulation.

## Code availability
The model was written in R[28] (several people were involved over a period of time, therefore several different versions of R were used), using primarily the functionality of the packages "data.table"[29] and "doParallel"[30]. Furthermore, "ggplot2"[31] was used for visualization. The code for popIBM is available from Zenodo[32].

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

## Acknowledgements

The authors wish to recognise the combined input from the "Expert group for mathematical modelling of SARS-CoV-2". Furthermore, we wish to recognise the hard work done by many people in many different ministries that help collect data on the fly in 2021 about the specific impacts of the many different restrictions. We also want to thank Peter Michael Bager (SSI) for a thorough read through of the article. The model was developed for the use in the expert group using a direct grant from the ministry of finance.

## Author contributions

KG devised and coded the popIBM model and all supporting functions, except for: activity matrices and adaptive behaviour devised and coded by LEC; vaccination rollout devised and coded by RSE. CK contributed to the work on activity matrices. All authors participated in writing the manuscript.

## Competing interests

The authors declare no competing interests.
