## [Peer Review File · Communications Medicine]

Reviewers' comments:

Reviewer #1 (Remarks to the Author):

Review of "Mass testing and local lockdowns effectively controls COVID-19" by Mr. Græsbøll and colleagues [Paper # COMMSMED-23-0629-T]

The research, titled "Mass testing and local lockdowns effectively controls COVID-19" investigate how different scenarios of mass testing and local lockdown interventions would have affected the daily hospital admissions and societal openness, applied to the Danish population. The study concludes that mass testing is an effective measure to reduce hospital admissions and prevent lockdowns.

The research questions this work investigates are important and relevant for the field. The work appears to be well thought through and important details of the population and population behavior are modeled. The model and the results add important and novel knowledge about mass testing and lockdowns as a tool for pandemic containment. The model is complex, and the manuscript contains many details on how the interventions are included in the model. However, a reorganizing of the Method section of the manuscript would be beneficial for the readers understanding of the work.

I advise the authors to revise the manuscript according to the comments below:

1. I find it difficult to get an overview/good understanding of how the mass testing and local lockdowns are implemented in the model and what aspects of the disease spread are affected by lockdown and testing. For instance, lockdowns change the activity matrix $A(t)$, meaning a reduction in contact between individuals, but what does mass testing affect in the model? The details are written, but scattered throughout the manuscript. A suggestion could be to include a visualization/figure of which part of the model mass testing and lockdowns affect, and/or restructure the Method section into separate subsections for the mass testing and local lockdowns.
2. What part of the model does poor adherence, or time varying adherence, to interventions affect, and how does this affect the results? This may be more important for other populations than the Danish but should for generalizability of the results be discussed in the Discussion section.
3. What were the requirements for a lockdown to be lifted? Was there a minimum number of days before it was lifted, or dependent on the incidence? Short and frequent lockdowns would be inconvenient for the population. This should be stated either in the Method section or in the Discussion.
4. Result section, line 120-132. These sentences are difficult to understand properly and should be reformulated. For instance, how does "difference in openness of 11" converts to "21 days of nationwide strict lockdown"? And the last two sentences is difficult to interpret.
5. On line 197, it is stated that all individuals with a positive test would be contacted by contact tracers. Did the contact tracing teams have capacity to contact everyone who had a positive test during the mass testing? The capacity of the contact tracing have been a problem in different countries during high incidence periods.

Reviewer #2 (Remarks to the Author):

General Comments:

The paper investigates the effectiveness of mass testing and local lockdown strategies in controlling the spread of COVID-19 during the Alpha-wave in Denmark in 2021. The study employs an individual-based simulation model to assess the impact of these strategies on reducing hospitalizations and maintaining societal freedom.

The article is well-written, featuring a concise and organized structure that effectively communicates the research objectives. The study's aims are clearly outlined, with a strong emphasis on evaluating the consequences of mass testing, local lockdowns, and their combination, comparing them to the real-world implementations in Denmark. The primary claims made in the study suggest that mass testing and local lockdowns in Denmark played a crucial role in averting a significant surge in hospital admissions, simultaneously allowing for greater societal freedom.

The results of the testing are presented in a comprehensible manner, offering both absolute counts and relative changes in hospitalizations. This presentation helps readers comprehend the specific context in Denmark and the potential implications for other countries.

In the section of the local lockdowns (lines 110-132), it might enhance clarity to provide a more detailed explanation. For instance, elaborating on how Figure 1B correlates with the differences in openness (0 and 11) in the two scenarios would be beneficial. Additionally, offering a clear explanation of how the value of 11/53% is related to the 21-day duration of a national lockdown would provide readers with a better understanding.

Major Comments:

- Line 37: Mention the date to avoid confusion regarding the simulation start date. Until line 232, I thought the simulation period started from 1st of January. I understand it is also mentioned on line 284.
- Lines 110-232: Introduce the concept of "openness" in the actual situation in the introduction, providing context, e.g., using Figure 1B to show the timeline of reopening schools and the initial 53% during lockdown.
- Lines 110-132: Clarify how Figure 1B translates to the difference in openness in the scenarios and explain the conceptual basis of the 21 days and 2.6 days. It is clear that the first part of the results in hospital admissions is based on cumulative numbers by summing all counts up, but a bit unclear to me how the difference of openness.
- Figure 1: Update y-axes and include the 53% initial value in the caption description.
- Table 1 (and Table A4): Specify that the numbers are medians, clarify that hospitalizations are total cumulative counts, and add the total counts of the observed scenario on the first row.
- Lines 137-138: Introduce cost in the introduction rather than presenting it without context in the last part of results.
- Line 176: Explain the difference or similarity between local and national lockdowns, as it's mentioned they are almost equivalent, but I cannot see how different or similar they are in the

results.

- Lines 253 and 263: Clarify the age groups used in the model and their relation to Figure A5. The model uses 0-9 years old (line 253) while 0-11 years old were not vaccinated (line 263). Also, are there 9 or 18 age groups in the model (line 253 vs 262)?
- Line 271 and 356: Explain the time delay between first and second doses and how VE changes after the second dose.
- Lines 356, 596: Use "vaccine effectiveness" (VE) instead of efficacy.
- Section A.3.1: Highlight the difference between true infection incidence and observed cases by testing. Similar in Figure A4, the y-axis shows test incidence and the caption use the fraction of test positivites.
- Figure A6: Explain why the thresholds of parishes are higher than those of municipalities and clarify the x-axis units (incidence counts or per population).

Minor Comments:

- Line 7: Specify "European Union (EU)" and maintain consistency in EU member states and EU members (Lines 64, 76, 403, 410).
- Line 251: Define "CPR" (possibly referring to the Danish Civil Registration).
- Line 312: Clarify whether "[0:1[" should be interpreted as 0 or 1.
- Line 323: Consider bolding "A(t)".
- Line 348: Specify that "rho_p" refers to each parish p.
- Line 366: If possible, provide a reference for the 50% asymptomatic probability.
- Line 404: Expand "ECDC" to "European Centre for Disease Prevention and Control."
- Line 405: Add the exact date until which the data was considered (e.g., "until July 4, 2021").
- Line 418: Specify the date or time when the exchange rate was used.
- Line 496: Expand "BBC" to "British Broadcasting Corporation."
- Line 501: Correct the reference citation format, e.g., "... by Klepac et al. [33] ..."
- Line 513: Correct the reference citation format, e.g., "... by BBC Pandemic study [32] ..."
- Line 559: Define "GDPR" (General Data Protection Regulation).
- Figure A5: Consider moving Figure A5 to a location after mentioning Table A1 to aid reader understanding.
- Figure A9: Include a legend for better understanding, indicating shaded areas as simulation results and solid lines as observed data.

Reviewer #1 (Remarks to the Author):

I advise the authors to revise the manuscript according to the comments below:

1. I find it difficult to get an overview/good understanding of how the mass testing and local lockdowns are implemented in the model and what aspects of the disease spread are affected by lockdown and testing. For instance, lockdowns change the activity matrix $A(t)$, meaning a reduction in contact between individuals, but what does mass testing affect in the model? The details are written, but scattered throughout the manuscript. A suggestion could be to include a visualization/figure of which part of the model mass testing and lockdowns affect, and/or restructure the Method section into separate subsections for the mass testing and local lockdowns.

We have now included a more detailed explanation of the effects of lockdown and testing in both the Introduction (L36-45) and end of the Methods Section (L511-521). We have considered making a figure but found it difficult to visualize as both lockdown and mass testing affect the same parameters (in- and out-going contacts for the individual) and may also affect an individual at any time, unrelated to disease state or progression of disease. The major difference is that mass testing prevents infection by reducing contacts of infected individuals, while lockdown prevents infection by reducing contacts of all individuals.

2. What part of the model does poor adherence, or time varying adherence, to interventions affect, and how does this affect the results? This may be more important for other populations than the Danish but should for generalizability of the results be discussed in the Discussion section.

We have added further discussion about adherence to interventions and the applicability in other countries in the Discussion section (L272-289).

3. What were the requirements for a lockdown to be lifted? Was there a minimum number of days before it was lifted, or dependent on the incidence? Short and frequent lockdowns would be inconvenient for the population. This should be stated either in the Method section or in the Discussion.

Thank you for pointing this out. We have added a mention of this in both the Method section (L519-521) and the Supplementary material to clarify the rules (A.6.1) including a reference. It was indeed a very interesting time to be in Denmark where restrictions frequently changed locally.

4. Result section, line 120-132. These sentences are difficult to understand properly and should be reformulated. For instance, how does “difference in openness of 11” convert to “21 days of nationwide strict lockdown”? And the last two sentences are difficult to interpret.

This conversion is central to the results of the manuscript, so thank you for pointing this out. We have reformulated this paragraph slightly, and inserted a paragraph that exemplifies these concepts to further understanding (L130-177, new paragraph 142-165)

5. On line 197, it is stated that all individuals with a positive test would be contacted by contact tracers. Did the contact tracing teams have capacity to contact everyone who had a positive test during the mass testing? The

capacity of the contact tracing have been a problem in different countries during high incidence periods.

Excellent question. The capacity of contact tracing changed over time in Denmark, and it is hard to give exact estimates of the efficiency, as the authority handling all contact tracing considered this personal information (GDPR protected), so no data was published to be studied, and all official webpages reporting efficiencies has in the meantime been taken down. We have unearthed an old news story from the national Danish broadcaster, which indicates that the number of contacts traced did only change little in the period (from 2.4 to 2.7 contacts traced). The capacity was also increased to be able to do contact tracing from 7.500 daily positive. We have added this information to the manuscript. (link: <https://www.dr.dk/nyheder/detektor/detektor-foer-jul-blev-smitteopsporingen-oversvoemmet-nu-har-de-tid-til-strikke>) (L251-256)

Reviewer #2 (Remarks to the Author):

Major Comments:

In the section of the local lockdowns (lines 110-132), it might enhance clarity to provide a more detailed explanation. For instance, elaborating on how Figure 1B correlates with the differences in openness (0 and 11) in the two scenarios would be beneficial. Additionally, offering a clear explanation of how the value of 11/53% is related to the 21-day duration of a national lockdown would provide readers with a better understanding.

As per request of both reviewers we have clarified the concept of “openness” by reformulating the paragraph slightly, and added a paragraph that exemplifies these concepts to further understanding (L130-177, new paragraph 142-165)

- Line 37: Mention the date to avoid confusion regarding the simulation start date. Until line 232, I thought the simulation period started from 1st of January. I understand it is also mentioned on line 284.

You are correct that this can be confusing. We have clarified throughout the manuscript that the study period starts January 11.

- Lines 110-232: Introduce the concept of "openness" in the actual situation in the introduction, providing context, e.g., using Figure 1B to show the timeline of reopening schools and the initial 53% during lockdown.

We have introduced ‘openness’ in the introduction (L74-79)

- Lines 110-132: Clarify how Figure 1B translates to the difference in openness in the scenarios and explain the conceptual basis of the 21 days and 2.6 days. It is clear that the first part of the results in hospital admissions is based on cumulative numbers by summing all counts up, but a bit unclear to me how the difference of openness.

As per request of both reviewers we have clarified the concept of “openness”. We have reformulated the paragraph slightly, and added a paragraph that exemplifies these concepts to further understanding (L130-177, new paragraph 142-165)

- Figure 1: Update y-axes and include the 53% initial value in the caption description.

We do not understand what the reviewer wanted to change in the y-axes (y-labels are displayed in the headings of the graphs, and [] are units), but we have specifically mentioned the 53% in the caption.

- Table 1 (and Table A4): Specify that the numbers are medians, clarify that hospitalizations are total cumulative counts, and add the total counts of the observed scenario on the first row.

We have added footnotes to the tables to clarify this.

- Lines 137-138: Introduce cost in the introduction rather than presenting it without context in the last part of results.

Cost estimated have been moved to the introduction. (L80-82)

- Line 176: Explain the difference or similarity between local and national lockdowns, as it's mentioned they are almost equivalent, but I cannot see how different or similar they are in the results.

We have reformulated the paragraph to better explain the likeness of local and national lockdowns. (L219-232) What we argue is that there would be no large visible difference in the results between national lockdown and all geographic areas in local lockdown, if those types of lockdowns had been implemented at the same point in time. However, there is no such direct comparison in the results. The closest we get to a direct comparison is that mid-May sees an approximate 75% lockdown of the country via local lockdowns in the orange scenario (Figure 1), which leads to a reduction in hospitalizations somewhat equivalent to the national lockdown in January. But it is difficult to make direct comparisons in the results as both season, variant proportion and vaccination uptake changes over time. The mathematical argument is that the dominant eigenvalue of the contacts and thus growth rate is identical for a full national lockdown and 100% of the country in local lockdown.

- Lines 253 and 263: Clarify the age groups used in the model and their relation to Figure A5. The model uses 0-9 years old (line 253) while 0-11 years old were not vaccinated (line 263). Also, are there 9 or 18 age groups in the model (line 253 vs 262)?

We have added a paragraph to clarify the difference between age groups and vaccinations target groups (L338-343). Essentially 17 vaccination target groups were defined by the Danish authorities (Table A1), and the model has one extra for the younger age group that was not included in the definition by the authorities. To make matters more confusing some of the vaccination groups are specific age groups (target groups 7-11), while other target groups have people of many ages (groups 1,2,4,5,6). Individuals in the model have both an age group and vaccination target group. There are 9 age groups and 18 vaccination target groups. However, not all combinations of age group and vaccination target group are represented in the model (because some vaccination target groups are age groups – so no 10-19 year olds can be in vaccination target group 8, which is for 75-79 year olds.), and for this reason Fig A2 (old figure number A5) is not fully populated.

- Line 271 and 356: Explain the time delay between first and second doses and how VE changes after the second dose.

In Denmark these were the recommended time between doses of the respective vaccine types. We have clarified this in the text (L348-357 + L438-445)

- Lines 356, 596: Use "vaccine effectiveness" (VE) instead of efficacy.

Thank you for pointing this out. We have changed it in the text.

- Section A.3.1: Highlight the difference between true infection incidence and observed cases by testing. Similar in Figure A4, the y-axis shows test incidence and the caption use the fraction of test positivities.

Thank you. We have clarified this in section A.3.1. We here assume that the reviewer refers to old Figure A3 / new figure A4 and the x-axis. (old Figure A4 / new figure A5 shows fraction of age group tested (test incidence), not test positive.) We have changed the x-axis label on old Figure A3 / new figure A4.

- Figure A6: Explain why the thresholds of parishes are higher than those of municipalities and clarify the x-axis units (incidence counts or per population).

The incidence limits were set by the authorities at the time, which we have now clarified in the text (L904-913). The reasoning behind higher limits in parishes is because they can have a very small population and are thus more sensitive to stochastic variation in number of infected cases. We have revised the caption and added units to the graph.

Minor Comments:

Thank you for providing these improvements to the manuscripts. We have revised accordingly.

- Line 7: Specify "European Union (EU)" and maintain consistency in EU member states and EU members (Lines 64, 76, 403, 410). Done

- Line 251: Define "CPR" (possibly referring to the Danish Civil Registration). Done (central person register)

- Line 312: Clarify whether "[0:1[" should be interpreted as 0 or 1. Done (in the range)

- Line 323: Consider bolding "A(t)". Done (here and in table A2)

- Line 348: Specify that " ρ_p " refers to each parish p. Done

- Line 366: If possible, provide a reference for the 50% asymptomatic probability. Done (in text and table A2)

- Line 404: Expand "ECDC" to "European Centre for Disease Prevention and Control." Done

- Line 405: Add the exact date until which the data was considered (e.g., "until July 4, 2021"). Done

- Line 418: Specify the date or time when the exchange rate was used. Done (the DKK is pegged to the Euro)

- Line 496: Expand "BBC" to "British Broadcasting Corporation." Done

- Line 501: Correct the reference citation format, e.g., "... by Klepac et al. [33] ..." Done

- Line 513: Correct the reference citation format, e.g., "... by BBC Pandemic study [32] ..." Done

- Line 559: Define "GDPR" (General Data Protection Regulation). Done

- Figure A5: Consider moving Figure A5 to a location after mentioning Table A1 to aid reader understanding.

Done (Figure A5 is now Figure A2)

- Figure A9: Include a legend for better understanding, indicating shaded areas as simulation results and solid lines as observed data. We have included similar wording in the caption.

REVIEWERS' COMMENTS:

Reviewer #1 (Remarks to the Author):

Thank you for revising the manuscript according to comments in the review.

I believe the manuscript reads better now and that the concept of openness and lockdown days are sufficiently explained.

I have no further comments to the manuscript.

Reviewer #2 (Remarks to the Author):

All the concerns have been addressed. I would recommend the paper for publication.